# Prevalence, Virulence Genes, Phylogenetic Analysis, and Antimicrobial Resistance Profile of *Helicobacter* Species in Chicken Meat and Their Associated Environment at Retail Shops in Egypt

**DOI:** 10.3390/foods11131890

**Published:** 2022-06-26

**Authors:** Amina Mohamed Elrais, Walid S. Arab, Khalid Ibrahim Sallam, Walaa Abd Elmegid, Fatma Elgendy, Walid Elmonir, Kálmán Imre, Adriana Morar, Viorel Herman, Haitham Elaadli

**Affiliations:** 1Department of Food Hygiene and Control, Faculty of Veterinary Medicine, Benha University, Benha 13511, Egypt; amenaelrayes_pfizer@yahoo.com (A.M.E.); walids444@gmail.com (W.S.A.); 2Department of Food Hygiene and Control, Faculty of Veterinary Medicine, Mansura University, Mansura 35516, Egypt; 3Department of Nutrition, Benha University Hospital, Benha University, Benha 13511, Egypt; aseel_soliman2011@yahoo.com; 4Department of Animal Wealth Development, Genetics and Genetic Engineering, Faculty of Veterinary Medicine, Benha University, Benha 13511, Egypt; fatma.algendy@fvtm.bu.edu.eg; 5Department of Hygiene and Preventive Medicine (Zoonoses), Faculty of Veterinary Medicine, Kafrelsheikh University, Kafrelsheikh 33516, Egypt; walid.elmonir@gmail.com; 6Department of Animal Production and Veterinary Public Health, Faculty of Veterinary Medicine, Banat’s University of Agricultural Sciences and Veterinary Medicine “King Michael I of Romania”, 300645 Timisoara, Romania; adrianamo2001@yahoo.com; 7Department of Infectious Diseases and Preventive Medicine, Faculty of Veterinary Medicine, Banat’s University of Agricultural Sciences and Veterinary Medicine “King Michael I of Romania”, 300645 Timisoara, Romania; viorel.herman@fmvt.ro; 8Department of Animal Hygiene and Zoonoses, Faculty of Veterinary Medicine, Alexandria University, Alexandria 21526, Egypt; haytham.kamal@alexu.edu.eg

**Keywords:** antibiogram, chickens, *Helicobacter pylori*, *Helicobacter pullorum*, virulence

## Abstract

*Helicobacter pylori* (*H. pylori*) and *Helicobacter pullorum* (*H. pullorum*) are frequently reported pathogens in humans and poultry, respectively. Nevertheless, the source of *H. pylori* is still unclear. This study aimed to detect *Helicobacter* spp. in chicken carcasses and to assess the antibiogram and the virulence genes of *Helicobacter* isolates. Three hundred chicken meat samples (100 each of chicken breast, liver, and gizzard), besides 60 swab samples from chicken processing surfaces, were collected from retail shops in Qalyubia Governorate, Egypt, and examined for the prevalence of *H. pylori* and *H. pullorum*. The 16S rRNA of three *H. pylori* and two *H. pullorum* isolates were sequenced to determine the genetic relationship between these two *Helicobacter* spp. Of the 300 chicken samples tested, 16 (5.33%) and 14 (4.67%) were positive for *H. pylori* and *H. pullorum*, respectively. Multiplex PCR revealed that the virulence genes vacuolating cytotoxin *A* (*vacA*)*s1*, cytotoxin-associated gene A (*cagA*), and restriction endonuclease-replacing gene A (*hrgA*) were detected in 66.7%, 77.8%, and 100% of *H. pylori* strains tested, respectively. *H. pylori* showed the highest resistance for clarithromycin, while *H. pullorum* exhibited the highest resistance towards erythromycin and ciprofloxacin. The study concluded that the chicken meat and giblets are potential sources of the virulent and antimicrobial-resistant strains of *H. pylori* of human origin.

## 1. Introduction

The poultry industry has been growing massively over the preceding two decades [1]. Chicken meat is considered one of the most commonly consumed food worldwide; thus, the hygienic procedures for providing chickens are tremendously related to public health and may be associated with many foodborne zoonotic pathogens of a substantial concern [2]. 

Meat-borne zoonotic diseases could be transmitted to humans by eating undercooked or inappropriately processed poultry meat. Furthermore, meat may become contaminated during poultry raising, handling, and slaughtering processes [3]. Among the major foodborne bacterial pathogens such as *Salmonella enterica*, *Yersinia enterocolitica*, *Campylobacter* spp., and Shiga toxigenic *Escherichia coli* (*E. coli*), *Helicobacter* spp. has been identified as an unusual infective agent. These enteric microorganisms are the most important causes of bacterial gastroenteritis, and also the most significant reasons for morbidity and mortality, especially in childhood [4,5]. As a result, *Helicobacter* spp. should be regarded as a hazardous foodborne pathogen.

*Helicobacter pylori* (*H. pylori*) is a microaerophilic, Gram-negative spiral bacterium, found in the stomachs of approximately half of the world’s population. It is strongly linked to peptic ulcer disease, duodenal ulcer, gastric adenocarcinoma, type B gastritis, and mucosa-associated lymphoid tissue (MALT) lymphoma [6,7,8]. A unique trait of *H. pylori* is its ability to colonize the gastric mucosa and thrive in the very acidic environment of the human stomach by producing huge amounts of urease enzyme, which raises the pH inside the stomach [9]. 

*Helicobacter pullorum* (*H. pullorum*) is a Gram-negative, microaerophilic, fastidious, slightly curved, non-spore-forming motile bacillus with monotrichous flagella [10]. *H. pullorum* is categorized as enterohepatic *Helicobacter* species [11]. This bacterium inhabits the intestinal tract of poultry and was initially detected in the liver and the duodenum of asymptomatic birds. It was also isolated from the liver and cecal contents of broiler chickens and laying hens suspected of vibrionic hepatitis [12,13]. In poultry slaughterhouses, *H. pullorum* has been found to be present on chicken carcasses, possibly due to its high concentration in the cecum and consequent contamination during slaughtering and evisceration [14]. Subsequently, the raw chicken meat may become contaminated. For this reason, the pathogenic potential role of this microorganism, as an emergent foodborne human pathogen, needs to be put into consideration. Moreover, *H. pullorum* was involved in several gastrointestinal disorders in humans, such as gastroenteritis, chronic liver disease, and inflammatory bowel disease [15,16,17].

*H. pylori* pathogenicity is linked to several virulence markers, including cytotoxin-associated gene A (*cagA*), vacuolating cytotoxin A (*vacA*), as well as restriction endonuclease-replacing gene A (*hrgA*) [18,19]. *CagA* is found in roughly half of all *H. pylori* strains and is responsible for the development of severe gastroduodenal disorders and gastric cancerous lesions [20]. Furthermore, *vacA* is present in all strains of *H. pylori* and can induce pores in the gastric host cells, resulting in the formation of vacuoles inside it [21]. In addition, *hrgA* is a novel potential virulence marker that was discovered during the characterization of the restriction-modification system (R-M) of *H. pylori* and was predominant among gastric cancer patients and *cagA*-positive *H. pylori* isolates [22]. 

Until now, treatment of *H. pylori* infections has been recommended by triple therapy, which is composed of clarithromycin, amoxicillin, or metronidazole, and a proton pump inhibitor [23]. The emergence of antibiotic-resistant *H. pylori* strains has become a global issue, and many published studies revealed that *H. pylori* strains isolated from food sources, as well as clinical specimens, show a high rate of resistance against various types of antimicrobial medications, including macrolide, fluoroquinolones, metronidazole, tetracyclines, penicillin, aminoglycosides, and sulfonamides [24,25,26]. 

In Egypt, although few studies concerning the occurrence of *H. pylori* [27,28] and *H. pullorum* [29] in chickens were reported, up to date, no study concerning the assessment of the sequence analysis and phylogenetic approach of *H. pylori* genotypes has been conducted. Therefore, the current research aimed to determine the prevalence of *H. pylori* and *H. pullorum* in chickens in Egypt, investigate the frequency of *vacA*, *cagA*, and *hrgA* virulence genes in *H. pylori* strains and evaluate the antimicrobial resistance phenotypes to clarithromycin, amoxicillin, metronidazole, tetracycline, and levofloxacin antibiotics in *H. pylori* and *H. pullorum* isolates, as well as to study the phylogenetic analysis and genetic relatedness of *H. pylori* and *H. pullorum* isolates.

## 2. Materials and Methods

### 2.1. Samples Collection

This study was conducted to investigate the prevalence and antibiotic resistance characteristics of *H. pylori* and *H. pullorum* in chicken meat, giblets and their associated environment, at retail shops in Qalyubia Governorate, Egypt. A total of 330 samples were collected from 10 retail shops of different sanitation levels distributed at a distance of 1–7 km in Benha and its suburbs during the period from February to April 2019. The 300 chicken samples included 100 breast fillets, 100 livers, and 100 gizzard samples. All samples were collected as freshly slaughtered chickens within 12 h from slaughter and were chilled at 4 °C. Additionally, 30 environmental pooled swab samples were collected from chicken processing surfaces at the retail shops as follows: 10 cutting board surfaces (3 swabs per board; 10 cm^2^ per swab), 10 knives (2 swabs per knife), and 10 workers’ hands (4 swabs per worker; 2 swabs per each hand) samples. The pooled swabs per sample were suspended in 10 mL buffered peptone water (BPW; Oxoid, Hampshire, UK). Samples were collected during 10 occasions of visits to the poultry retail shops and were transported in a cold icebox to the laboratory to be tested within 1 h from arrival.

### 2.2. Isolation and Identification of Helicobacter *spp.*

For selective pre-enrichment, 25 g of each sample (chicken meat, liver, or gizzard), were added to 225 mL of a Brucella broth supplemented with 5% sheep defibrinated blood, and DENT selective supplement (Oxoid, Hampshire, UK), and the mixture was homogenized in Stomacher^®^ 400 (Seward, Worthing, UK). The mixture was then divided into two 250 mL flasks and incubated at 37 °C for 48 h under a microaerophilic condition using BBL GasPak™ jars (Becton Dickinson, Franklin Lakes, NJ, USA), supplemented with CampyGen bags (Oxoid, Hampshire, UK). After incubation, 100 µL of the mixture were inoculated onto Columbia blood agar base (Oxoid, Hampshire, UK), supplemented with 5% sheep blood and DENT selective supplements. The plates were incubated for up to 7 days at 37 °C under a microaerophilic condition, as previously described. Suspected colonies were further identified using Gram’s staining (Gram-negative for *Helicobacter* spp.), oxidase (positive for *Helicobacter* spp.), urease (positive for *H. pylori*), and nitrate reduction (positive for *H. pullorum*) tests.

### 2.3. Molecular Confirmation of Helicobacter *spp.*

Specific primer sets (Table 1) were synthesized by Metabion, Steinkirchen, Germany to be used for the amplification of the 16S rRNA of *Helicobacter* genus and *H. pullorum*, as well as the specific primers of *H. pylori*-specific phosphoglucosamine mutase gene (*glmM*), and virulence factors including *hrgA*, *cagA*, and *vacA* genes of for *glmM* gene.

In summary, a QIAamp DNA Mini kit (Qiagen, Hilden, Germany) was used to extract DNA from a pure culture. For genus confirmation through 16S rRNA amplification, 5 µL of DNA template were mixed with 12.5 µL of EmeraldAmp Max PCR Master Mix (Takara Bio, Kusatsu, Japan), 1 µL of each primer (20 pmol), and 5.5 µL of water. The PCR mix was transferred to theApplied Biosystems 2720 Thermal Cycler (Applied Biosystems, Foster City, CA, USA), under the following conditions: 94 °C, for 5 min, followed by 35 cycles of 94 °C for 1 min, 54 °C for 1 min, and 72 °C for 1 min, and finally, an extension for 10 min at 72 °C. The *H. pylori* ATCC 43,504 strain and distilled water were used as positive and negative controls, respectively.

For species identification, primers targeting the *glmM* gene and *H. pullorum*-specific 16S rRNA were used for the identification of *H. pylori* and *H. pullorum*, respectively. The PCR mix and cycling conditions were the same as previously described in genus-specific *16S rRNA* except for annealing temperatures.

A multiplex PCR was used to detect the virulence genes *hrgA, cagA, and vacA* in *H. pylori* isolates as previously described [32]. The PCR mixture included a 5 µL DNA template, 12.5 µL of EmeraldAmp Max PCR Master Mix (Takara Bio), 1 µL of each primer (10 pmol), and water up to 25 µL. The PCR cycling conditions were identical to those previously described in genus-specific 16S rRNA except for the annealing temperature, which was set at 52 °C for 1 min. The DNA of *H*. *pylori* was used as a positive control, while the DNA isolated from *E*. *coli* K12DH5α served as a negative control. PCR products were electrophoresed in a 1.5% agarose gel with 0.3% ethidium bromide in a 10% Tris–borate–EDTA buffer. DNA signals in the gel were visualized under a UV transilluminator.

### 2.4. Antimicrobial Susceptibility Testing

The agar dilution model was used to evaluate antibiotic susceptibility profiles, according to the Clinical and Laboratory Standards Institute guidelines [33]. The *Helicobacter* isolates were collected from 72 h culture on Blood agar and suspended in saline, to reach the 2.0 McFarland opacity standard. Then, 2 µL of each suspension were spot inoculated in Mueller-Hinton agar (Oxoid, Hampshire, UK), supplemented with 5% sheep blood, and serial two-fold dilutions of eight antibiotics (Sigma-Aldrich, St. Louis, MO, USA). The inoculated plates were incubated under microaerophilic conditions, at 37 °C for 72 h, as previously described. For *H. pylori*, the antibiotic resistance breakpoints for amoxicillin, metronidazole, tetracycline, and levofloxacin were determined according to the European Committee on Antimicrobial Susceptibility Testing [34], while the breakpoint for clarithromycin was adopted from CLSI [33]. For *H. pullorum*, the breakpoints for ampicillin, erythromycin, tetracycline, and ciprofloxacin were assumed according to *Campylobacter* and related species [29,35,36]. For quality control, *H. pylori* ATCC 43504 reference strain was used for *H. pylori* isolates. For *H. Pullorum* isolates, *C. jejuni* ATCC 33560 was used as a control for ciprofloxacin, erythromycin, and tetracycline, while *Staphylococcus aureus* ATCC 43300 was used as a control for ampicillin.

### 2.5. Helicobacter Species 16S rRNA Gene Sequencing and Phylogenetic Analysis

Using the QIAquick gel extraction kit (Qiagen, Valencia, CA, USA), the 16*S rRNA* PCR products of three *H. pylori* isolates (one isolate from each of meat, liver, and environmental swab) and two *H. pullorum* isolates (one isolate from each of meat and liver samples) were purified. The purified products were sequenced in both directions, using a Big-Dye Terminator v3.1 cycle sequencing kit (Applied Biosystems, Foster City, CA, USA) in an Applied Biosystems 3130 genetic analyzer (Applied Biosystems), according to the manufacturer’s instructions. The BLAST 2.2 program (National Center for Biotechnology Information; NCBI) was used to confirm the nucleotide sequence identity. The phylogenetic tree was generated using the MegAlign module of DNASTAR Lasergene software V.12.1 [37], and phylogenetic analyses were performed in MEGA6 software using maximum likelihood, neighbor-joining, and maximum parsimony [38].

### 2.6. Statistical Analysis

Fisher’s exact test was applied to determine the significant difference between the prevalence of *Helicobacter* species in chicken meat and their associated environment. Significance was determined at *p* < 0.05.

## 3. Results and Discussions

### 3.1. Prevalence of Helicobacter *spp.* in Chicken Meats and Swab Samples

Livestock (particularly poultry) is considered a crucial reservoir of many pathogenic microorganisms. *Helicobacter* has recently developed a public health concern as an emerging foodborne pathogen [4]. For that reason, the microbial quality assessment of chicken meat is important to reduce the load of *Helicobacter* in meat. Information regarding chickens as an essential reservoir for *H. Pylori* dissemination to humans is very limited [28]. In the current study, *H. pylori* was detected in 4% (4/100) of chicken breast meat samples. In chicken liver and gizzard samples, however, *H. pylori* was isolated from 10% (10/100) and 2% (2/100) samples, respectively (Table 2). The overall prevalence of *H. pylori* among the 300 broiler chicken samples (meat and giblets) was 5.33% (16/300) (Table 2). This result was consistent with that of Dairouty et al. [27], who revealed that 5% (1/20) of raw poultry meat were positive for *H. pylori*. Conversely, a much higher prevalence rate of 36% (4/11) was reported for *H. pylori* by Meng et al. [39], in fresh raw chickens. *H. pylori* contamination in the chicken meat samples tested in the present study could be attributed to the contaminated hands of the butchers, veterinarians, and slaughterhouse workers during handling, chicken portions, and giblets preparation and packaging. Besides, usage of unclean water during the washing of the chicken carcasses could be another potential reason for the presence of *H. pylori* in the chicken meats. Furthermore, the incidence of *H. pylori* in chicken specimens may be due to cross-contamination from knives or other slaughterhouse equipment.

Regarding the prevalence of *H. pullorum* in chicken meat and giblets, *H. pullorum* was detected in 2% (2/100) meat samples. In chicken liver and gizzard samples, however, *H. pullorum* was detected in 6% (6/100) and 6% (6/100) of the samples tested, respectively. The overall prevalence of *H. pullorum* among the 300 broiler chicken samples (meat and giblets) was 4.67% (14/300) (Table 2)**.** A lower prevalence rate was reported by Gholami-Ahangaran et al. [40], who indicated that *H. pullorum* was present in 2% (2/100) of the examined liver samples. On the contrary, higher prevalence rates of 23.5% (4/17) and 24% (12/50) were recorded for *H. pullorum* in raw fresh chicken meat [4] and thigh chicken samples [41], respectively. The presence of *H. pullorum* in chicken meat samples could be attributed to its dissemination from the poultry cecum and consequent contamination of chicken carcasses during poultry processing, and this suggests that this organism may be a potential risk factor for zoonotic foodborne transmission to human consumers. Furthermore, the isolation of *H. pullorum* from the chicken liver could be attributed to the bacterium’s ability to enter the liver via retrograde transfer from the duodenum. *H. pullorum* may also translocate from the gut lumen to the portal circulation [42].

Concerning the presence of *H. pylori* and *H. pullorum* in environmental swab samples, just two *H. pylori* isolates were isolated from two different cutting boards, while *H. pullorum* was only detected in one environmental sample, specifically from a worker’s hand (Table 2). This finding may be due to infected workers’ hands and infrequent cleaning and disinfection of the cutting boards before use.

The frequencies of the different types of *Helicobacter* (*pylori*, *pullorum*, and others) were compared among retail chicken organs (breast meat, liver, and gizzard) and a significant difference were detected at *p* < 0.05, with the highest frequency noticed for *H*. *pylori* in liver, while there was no significant difference between the frequencies of different types of *Helicobacter* (*pylori*, *pullorum*, and others) in the environmental swap samples (cutting board, knives, and working hands).

### 3.2. Phylogenetic Analysis of Partial 16S rRNA Gene Sequencing of Helicobacter Species

The PCR amplicon for the *16S rRNA* gene of *Helicobacter* spp. Was detected at the particular expected size of 398 bp (Figure 1A,B). Additionally, *H*. *pullorum*-specific 16S rRNA was detected at the expected size of 447 bp (Figure 2A), while the PCR product specific of *glmM* gene-specific for the characterization of *H. pylori* was detected at the expected size of 294 bp (Figure 2B).

The resultant 16S rRNA gene sequences of the selected three *H*. *pylori* and two *H*. *pullorum* in this study were submitted to the GenBank nucleotide database, under the following accession numbers MW404637, MW404633, and MW407986 for *H*. *pylori* isolates and MW407962 and MW404621 for *H*. *pullorum* isolates (Table 3).

The 16S rRNA gene sequencing of the selected three *H*. *pylori* indicated that two isolates were clustered together with a 100% genetic similarity and were supported with a bootstrap value of 98%, but the third one was distantly related with a 99.2% sequence similarity. On the other hand, the nucleotide sequences of the 16S rRNA *gene* from the selected two *H*. *pullorum* isolates were located in the same cluster with a 100% genetic identity. It was also found that these two *H*. *pullorum* isolates of the current study were highly related to the other *H*. *pullorum* sequences, which were retrieved from the NCBI GenBank databases. It means that *H*. *pullorum* isolates had a very low genetic diversity. The identities of the *H*. *pylori* and *H*. *pullorum* isolates are shown in Figure 3.

The homology search (BLAST) of the sequenced *16S rRNA* gene was conducted for determining their identities and knowing their phylogeny trees. The phylogenetic analysis of the three *H*. *pylori* sequences in the current study showed high genetic identity to all the retrieved *H*. *pylori* sequences, which were of human origin, and this finding emphasized that humans could be the main source of *H*. *pylori* contamination in chicken meat and livers. Furthermore, the phylogenetic analysis of the two *H*. *pullorum* sequences of the present study revealed a 100% homology to one isolate (L36145), which was obtained from a human patient with gastroenteritis, and also showed a 99.8% identity to another isolate (AY394474), which was recovered from the cirrhotic liver of human with hepatitis C. This finding indicated that *H*. *pullorum* could be a potential foodborne zoonotic pathogen. The phylogenetic analysis of the *H*. *pylori* and *H*. *pullorum* isolates is shown in Figure 4.

The phylogenic diversity among the analyzed *Helicobacter* isolates might be attributed to the difference in sampling sites from long-distance localities in Qalyubia Governorate from which chicken meat and giblets were collected, and hence, a variation of the strains isolated from these different samples. On the other hand, the close identity between the *16S rRNA* sequences of the strains analyzed could be attributed to the cross-contamination of the chicken samples.

### 3.3. Genotypic Characterization of H. pylori Virulence Genes

The multiplex PCR verified the presence of the virulence genes *vacA*, *cagA*, and *hrgA* at the expected molecular size of 259, 499, and 594 bp, respectively, in the *H*. *pylori* isolates (Figure 5).

The frequency distributions of *vacA*, *cagA*, and *hrgA* virulence genes of *H. pylori* isolates in this study are shown in Table 4. Based on the molecular analysis, *vacA*s1, *cagA*, and *hrgA* were detected in 66.7% (12/18), 77.8% (14/18), and 100% (18/18) of the 18 *H. pylori* isolated strains, respectively. Accordingly, the most common virulence marker was the *hrgA* gene. Additionally, the highest frequency of the tested virulence genotypes was detected in the chicken isolates from the liver and breast meat of broiler chickens which are commonly consumed food for humans.

The existence of *vacAs1* and *cagA* genes among the isolates in the present investigation was in agreement with that of Hamada et al. [28], who found that *vacA* and *cagA* genes were detected in 57.1% and 42.9% of *H. pylori* isolates recovered from 90 chicken meat, liver, and gizzard samples. Similarly, Hemmatinezhad et al. [43] reported that *vacA* and *cagA* were the most commonly identified genes in the *H. pylori* isolates from ready-to-eat foods since they existed in 78.37% and 41.89% of the tested isolates. On the other hand, a much lower existence rate of 20% was determined for both *vacAs1* and *cagA* in *H. pylori* isolates from minced meat samples [44]. Likewise, a considerable prevalence of *vacAs1* and *cagA* virulence factors were shown previously in many foods of animal origin [45,46].

The existence of the virulence genes in *H*. *pylori* evoked adverse effects on human consumers. The presence of *vacAs1 and cagA* virulence factors enables the colonization and survival of *H. pylori* within the gastric mucosa through complex mechanisms, such as adhesion to gastric epithelial cells, interleukin-8 production, stimulation of inflammatory response, formation of intracellular vacuoles, induction of apoptosis of gastric epithelial cells, and lastly gastritis, gastroduodenal ulcers, and even gastric cancer in individuals who eat these examined contaminated chicken meat samples [47]. In addition, a significant relationship between *vacAs1* expression and peptic ulcer disease (PUD) has previously been found [18,48].

Concerning the *hrgA* gene, the current study is considered the first record of its detection in *H. pylori* from a food source. Moreover, it had the highest frequency (100%) in the studied *H. pylori* isolates of chicken meat, giblets, and swab samples. It is a novel gene that was identified during the examination of the hpyIIIR−hpyIIIM locus in Western and Asian *H. pylori* strains, and this gene was found in place of the hpyIIIR gene and located upstream of hpyIIIM in 34% (70/208) of the examined strains. From this time, it is named restriction endonuclease-replacing gene A [22]. This recently identified gene was more predominant in western countries than in Asia and more prevalent in gastric cancer patients in comparison to in patients without gastric cancer in East Asian countries. In addition, it was more abundant in *cagA^+^* than *cagA^−^* isolates in western strains [49]. The virulent role of *hrgA* was also investigated, and it was observed that it had a direct function in interleukin-8 induction and the apoptosis of gastric epithelial cells [22]. Thus, our report suggests that *hrgA* could potentially be a public health hazard for humans who consumes chicken meat contaminated with *hrgA^+^ H. pylori* strains.

### 3.4. Antimicrobial Resistance Profiles of the H. pylori and H. pullorum Isolates

The antibacterial agents that were selected for evaluation are commonly used in human medicine to treat patients suffering from *Helicobacter* infection. In this study, four *H. pylori* isolates (44.4%) were resistant to at least one antibiotic (Table 5), while two isolates showed multiple drug resistance (MDR) to more than three classes of antibiotics. The highest resistance rates of 44.4% and 33.3% were for clarithromycin and metronidazole, respectively, while the lowest rate was determined for amoxicillin (11.1%) (Table 5).

In the case of *H. pullorum*, six isolates were resistant to at least one antibiotic (85.7%), and three isolates (42.9%) were MDR (Table 5). All isolates were susceptible to ampicillin but showed high resistance rates for erythromycin (85.7%) and ciprofloxacin (71.4%).

In respect of *H. pylori* isolates, we found that *H. pylori* strains exhibited a high rate of resistance toward clarithromycin, metronidazole, tetracycline, levofloxacin, and amoxicillin. Our findings were consistent with those of Hamada et al. [28], who found high levels of *H. pylori* resistance to amoxicillin, penicillin, oxytetracycline, nalidixic acid, ampicillin, and norfloxacin in chicken meat, liver, and gizzard; Mashak et al. [50] who reported that *H. pylori* strains were resistant towards tetracycline, erythromycin, levofloxacin, and amoxicillin in raw meat samples; Gilani et al. [44], who found that *H. pylori* bacteria displayed a high resistance against ampicillin, erythromycin, amoxicillin, tetracycline, and clarithromycin in meat product samples; Mousavi et al. [44], who stated that *H. pylori* isolates from milk displayed strong resistance toward ampicillin, tetracycline, erythromycin, and metronidazole; and Ranjbar et al. [46], who mentioned that *H. pylori* strains from traditional dairy products harbored a high incidence of resistance against ampicillin, amoxicillin, tetracycline, erythromycin, and metronidazole. Besides, previous studies reported by Secka et al. [51] and Yahaghi et al. [52] revealed that *H. pylori* isolated from food specimens regularly showed an incidence of resistance against amoxicillin, metronidazole, ampicillin, and oxytetracycline. In addition, epidemiological studies conducted in China, Taiwan, the Kingdom of Saudi Arabia, Egypt, Nigeria, Iran, India, Brazil, Argentina, and Colombia revealed that *H. pylori* strains obtained from medical samples are highly resistant to amoxicillin, metronidazole, quinolones, and tetracycline [53,54].

The emergence of multidrug-resistant foodborne bacterial pathogens in humans is mainly attributed to the frequent misuse and overuse of antibiotics for prophylaxis and growth promotion in the poultry industry. Additionally, the abuse of antimicrobial agents and self-medication by human beings may be considered an additional source. It is likely that the *H. pylori* bacteria can be transmitted from infected butchers and workers to meat samples through cross-contamination during meat handling in poultry slaughterhouses. Our finding of the antibiotic resistance pattern of *H. pylori* isolates revealed that the meat, livers, and gizzards of the examined poultry may be a possible vehicle for antibiotic-resistant *H. pylori* and subsequently could be hazardous to human health.

Concerning *H. pullorum* strains, three *H. pullorum* isolates collected in this study showed high resistance rates for erythromycin, ciprofloxacin, and tetracycline (42.9–85.7%), which agreed with other studies [4,36,41]. All *H. pullorum* isolates were sensitive to ampicillin, which was in line with the results of Zanoni et al. [36] and Hassan et al. [29]; nonetheless, Ceelen et al. [15] showed a higher rate of ampicillin resistance among poultry isolates in Europe. This observed incidence of antimicrobial resistance in chicken samples was attributable to the frequent use of quinolones and tetracyclines for prophylaxis and growth promotion in the Egyptian poultry industry.

## 4. Conclusions

The present study concluded that approximately 5% of chicken meat and giblets marketed in Egypt were contaminated with *H. pylori* and/or *H. pullorum* which are resistant to the antimicrobials recommended for human treatment, and hence, the consumption of undercooked chicken meat and giblets is considered a potential public health hazard to humans. Contamination of chicken carcasses can occur during slaughtering and/or processing or when they come into contact with contaminated hands or contaminated water in poultry abattoirs. It is critical for slaughterhouses to maintain good hygienic measures and sanitary practices to limit contamination of meat with *H. pylori* and *H. pullorum*. The present study also found a very high (100%) frequency of the *hrgA* gene in the isolated *H. pylori* strains. In addition, more than 20% of *H. pylori* isolates were resistant to three or more antibiotics. This highlights the significance of evaluating the antibiotic susceptibility profile of *Helicobacter* infections to recommend using the most effective antimicrobial agents for its eradication.

## Figures and Tables

**Figure 1 foods-11-01890-f001:**
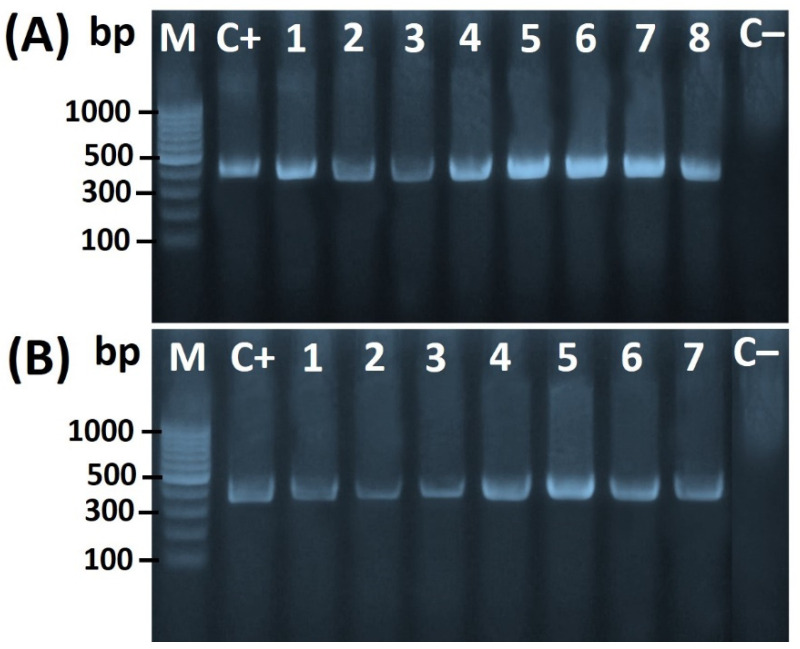
(**A**) *H. pylori* (lanes 1–8). (**B**) *H. pullorum* (lanes 1–7). Agarose gel electrophoresis showing *16S rRNA* PCR amplicon (398 bp) for *Helicobacter* genus. Five microliters from the PCR product were separated by electrophoresis on a 1.5% agarose gel and visualized under UV light. M: DNA marker (Gene Ladder 100) used as a reference for a fragment size; Lane C+: positive control from *H. pylori* ATCC 43,504 strain. C−: negative control of *Escherichia coli* (*E*. *coli*) K12 DH5α as a negative control.

**Figure 2 foods-11-01890-f002:**
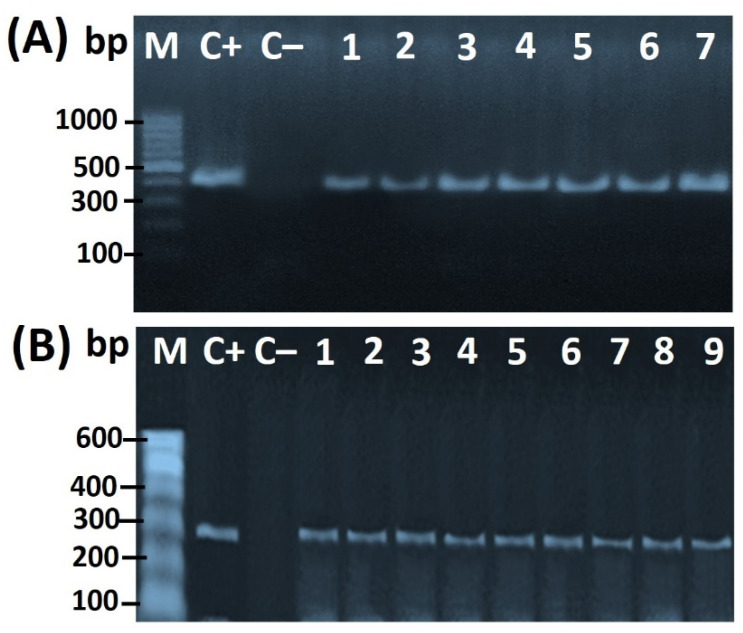
(**A**) Agarose gel electrophoresis of PCR amplicon (447 bp) of the *16S rRNA* specific for *H*. *pullorum* strains (lanes 1–7). (**B**) Agarose gel electrophoresis of PCR product of *glmM* gene (294 bp) specific for the characterization of *H. pylori* strains (lanes 1–9). Lane M: 100 bp ladder as a molecular-size DNA marker. Lane 1: *H. pylori* positive control for the *glmM* gene. Lane 2: negative control from *E*. *coli* K12DH5α. Lanes from 1 to 9: positive *H. pylori* strains.

**Figure 3 foods-11-01890-f003:**
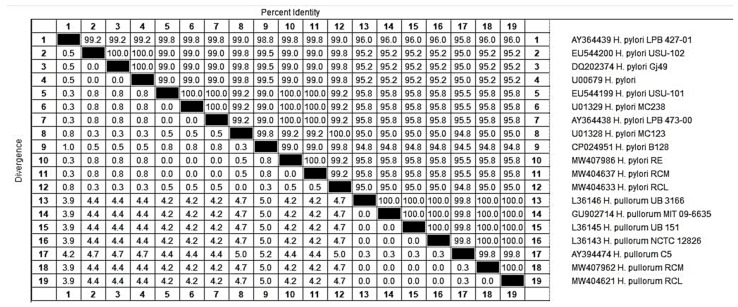
Identity percentages detected through the homology search (BLAST-N) of the *16S rRNA* sequences of the isolated *H*. *pylori* and *H*. *pullorum.*

**Figure 4 foods-11-01890-f004:**
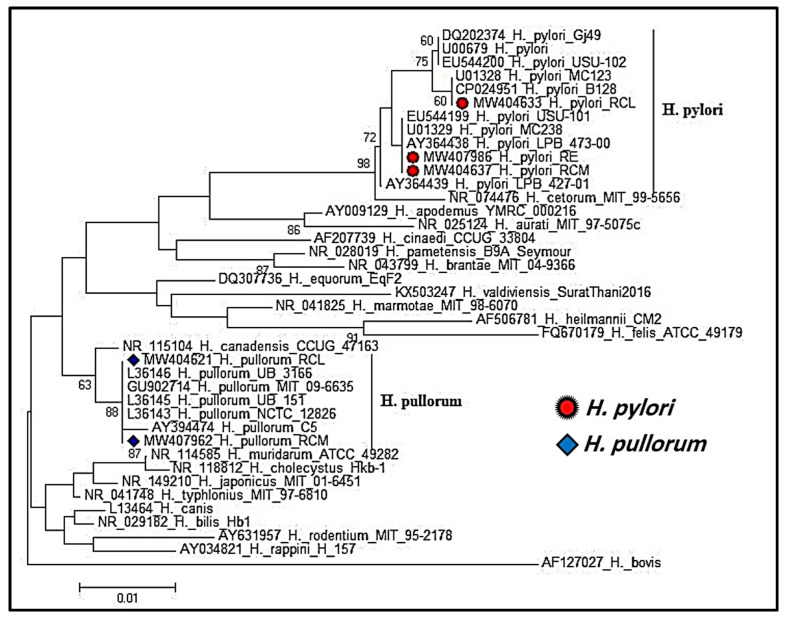
Phylogeny analysis of the *16S rRNA* sequences of the isolated three *H*. *pylori* (a red circular shape) and the two *H*. *pullorum* (a blue rhombus shape) isolates.

**Figure 5 foods-11-01890-f005:**
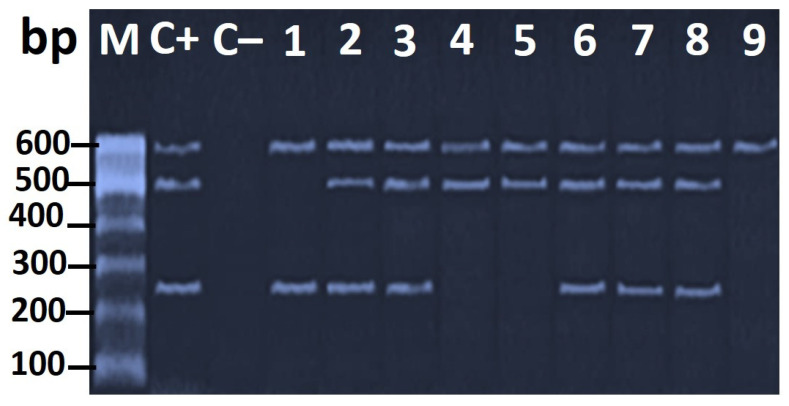
Agarose gel electrophoresis of the multiplex PCR of *vacA* (259 bp), *cagA* (499 bp), and *hrgA* (594 bp) as virulence genes of *H. pylori* strains. Lane M: 100 bp ladder as a molecular-size DNA marker. Lane C+: *H. pylori* ATCC 43504 strain positive control. Lane C−: *E*. coli K12 DH5α negative control. Lanes 2, 3, 6, 7, and 8: *H. pylori* positive control for *vacA*, *cagA*, and *hrgA* genes. Lanes 4 and 5: positive control of *H. pylori* for *cagA* and *hrgA* genes. Lane 1: positive control of *H. pylori* for *vacA* and *hrgA* genes. Lane 9: positive control of *H. pylori* for the *hrgA* gene. Lanes 1 and 2 represent chicken meat; lanes 3 and 4 represent chicken gizzard; lanes 5 to 9 represent chicken liver.

**Table 1 foods-11-01890-t001:** Primers used in this study.

Target Gene	Primers Sequences (5′–3′)	Product Size (bp)	Reference
*Helicobacter* spp. 16S rRNA	5′-AAGGATGAAGCTTCTAGCTTGCTA-3′ 5′-GTGCTTATTCGTGAGATACCGTCAT-3′	398	Tabrizi et al. [30]
*Helicobacter pullorum* (*H. pullorum*)*-specific* 16S rRNA	5′-ATG AAT GCTAGTTGTTGTCAG-3′ 5′-GATTGGCTCCACTTCACA-3′	447	Stanley et al. [10]
Helicobacter pylori (*H. pylori*)-specific phosphoglucosamine mutase gene (*glmM*)	5′-GAATAAGCTTTTAGGGGTGTTAGGGG-3′ 5′-GCTTACTTTCTAACACTAACGCGC-3′	294	Safaei et al. [31]
Restriction endonuclease-replacing gene A (*hrgA*)	5′-TCTCGTGAAAGAGAATTTCC-3′ 5′-TAAGTGTGGGTATATCAATC-3′	594	Tiwari et al. [32]
Cytotoxin-associated gene A (*cagA*)	5′-GCGATTGTTATTGTGCTTGTAG-3′ 5′-GAAGTGGTTAAAAAACAATGCCCC-3′	499	
Vacuolating cytotoxin A (*vacA*)	5′-ATGGAAATACAACAAACACAC-3′ 5′-CTGCTTGAATGCGCCAAAC-3′	259	

**Table 2 foods-11-01890-t002:** Prevalence of *Helicobacter* species in chicken meat and their associated environment at the retail shops.

Source	Sample Type	*Helicobacter* spp.						Total	
		*H. pylori*		*H. pullorum*		Others			
		No.	%	No.	%	No.	%	No.	%
Retail chicken (*n* = 300)	Breast meat (100)	4	4	2	2	0	0	6	6
	Liver (100)	10	10	6	6	4	4	20	20
	Gizzard (100)	2	2	6	6	2	2	10	10
Environmental swabs (*n* = 30) *	Cutting boards (10)	2	20	0	0	0	0	2	20
	Knives (10)	0	0	0	0	0	0	0	0
	Workers’ hands (10)	0	0	0	0	1	10	1	10
Total	330	18	5.45	14	4.24	7	2.12	39	11.82

* Each sample is a pool of 2–4 swabs.

**Table 3 foods-11-01890-t003:** The accession numbers of 16S rRNA gene sequence of the selected five *Helicobacter* species.

Gene	Isolate spp.	Isolate ID	Source of Isolates	Accession Number
16S rRNA	*H. pylori*	*H. pylori*_RCM	Chicken meat	MW404637
		*H. pylori*_RCL	Chicken liver	MW404633
		*H. pylori*_RE	Retail shop environment (Cutting board swabs)	MW407986
	*H. pullorum*	*H. pullorum_*RCM	Chicken Meat	MW407962
		*H. pullorum*_RCL	Chicken Liver	MW404621

**Table 4 foods-11-01890-t004:** Frequency distributions of the virulence genes in *H. pylori* isolates from chicken meat and environmental samples.

Source	Number of Isolates	*vacAs1*		*cagA*		*hrgA*	
		Number	%	Number	%	Number	%
Breast	4	4	100	2	50	4	100
Liver	10	6	60	8	80	10	100
Gizzard	2	0	0	2	100	2	100
Environment	2	2	100	2	100	2	100
Total	18	12	66.7	14	77.8	18	100

**Table 5 foods-11-01890-t005:** Antimicrobial resistance profiles of *H*. *pylori* and *H*. *pullorum* isolates from chicken meat and environmental samples.

Isolates	Antibiotic	Bp	Isolates Number According to the Results of MIC (µg/mL)														ABR No. (%)	MDR No. (%)
			˂0.12	0.12	0.25	0.5	1	2	4	8	16	32	64	128	256	˃256		
*H. pylori* (*n* = 18)	Amoxicillin	˃0.125	14	2					2								2 (11.1)	2 (22.2)
	Clarithromycin	≥1	2	2	6			2			2	2		2			8 (44.4)	
	Metronidazole	˃8			3	1	1	3	4		4		2				6 (33.3)	
	Tetracycline	>1		4	3	7		2		2							4 (22.2)	
	Levofloxacin	>1	4	3	3	4			2			2					4 (22.2)	
*H. pullorum* (*n* = 14)	Ampicillin	≥32			2	3	7		2								0 (0)	3 (42.9)
	Erythromycin	≥8							2		1	3	2	3	3		12 (85.7)	
	Tetracycline	≥16							3	5	3		1			2	6 (42.9)	
	Ciprofloxacin	≥4					1	3	3	1	2	4					10 (71.4)	

Bp: breakpoints for antibiotic resistance; ABR: antibiotic resistance isolates; MDR: multiple drugs-resistant isolates (≥3 classes of antibiotics); No.: isolates number.

## Data Availability

Data is contained within the article.

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
