# Peer review of "Prevalence, Virulence Genes, Phylogenetic Analysis, and Antimicrobial Resistance Profile of Helicobacter Species in Chicken Meat and Their Associated Environment at Retail Shops in Egypt"

_foods, 2022, doi:10.3390/foods11131890_

Round 1

Reviewer 1 Report

This study was conducted to determine the contamination frequency of Helicobacter species in chicken meat in a specific area. The objective and experimental design of this study are reasonable, and the description of obtained results is appropriate. Thus, this reviewer suggests some minor comments as follows.

L94-96 Please double-check grammar errors.

L105-117 If possible, it would be better to mention how different samples represent different slaughterhouse distributions since slaughterhouses have a high incidence of microbial contamination.

L194 It would be better to specify “chicken meat quality”, since it generally means the quality attributes related to consumer acceptance.

L198 10% and 2? the unit should be consistent. 

Author Response

  • L94-96 Please double-check grammar errors. Checked and corrected.
  • L105-117 If possible, it would be better to mention how different samples represent different slaughterhouse distributions since slaughterhouses have a high incidence of microbial contamination. a clause "different poultry ships of different sanitation levels" clarifying such meaning is added in the revised manuscript.
  • L194 It would be better to specify “chicken meat quality”, since it generally means the quality attributes related to consumer acceptance. It is specified in the revised manuscript as "the microbial quality assessment of chicken meat".
  • L198 10% and 2? the unit should be consistent. Corrected. (%) is added.

            Thank you so much for your revision and valuable corrections

Reviewer 2 Report

The current study investigated the prevalence of Helicobacter pylori and Helicobacter pullorum among 330 chicken and environment samples at retail shops. Virulence genes and antimicrobial susceptibility of 18 isolated strains were further confirmed and the 16S rRNA gene sequencing and phylogenetic analysis were also performed. The results got from the current study may provide some basic information for the effective control of the Helicobacter pylori and Helicobacter pullorum in the poultry industry.

Line 108  More information about the ten retail shop should be provided such as the scale and the distance between each other. This will help a good understanding of the results.

Line 203-208 Analysis should base on the results from the results that had already been obtained such as the difference prevalence among different sample types but not the deduction which cannot be proved by the results.

Line 200 Table 2 Nonparametric statistical tests such as the Chi-square test or T-test should be performed.

Line 224 Reference should be added.

Line 226-227 H. pullorum was not detected in the environment swabs but the “others” do.

Line 229-231 The result is not detailed enough to sustain this argument unless statistical analysis or molecular traceability were performed.

Line 260-261 “It was also found that these two H. pullorum isolates of the current study……” which two?

Fig.4 The strains in the current study should be emphasized to get a good understanding of the results.

Line 280 Phylogeny analysis and discussion should also focus on the distance of strains isolated from different sample types. This may provide some information on the cross-contamination mentioned before.

Line 370-374 The frequent use of antibiotics for prophylaxis and growth promotion in the poultry industry may be the main reason but not the translation from human staff.

Author Response

  • Line 108  More information about the ten retail shops should be provided such as the scale and the distance between each other. This will help a good understanding of the results. The required details are added in the revised manuscript (Line 108-109).
  • Line 203-208 Analysis should base on the results from the results that had already been obtained such as the difference prevalence among different sample types but not the deduction which cannot be proved by the results. An appropriate analysis is written and the sentence is corrected in the revised manuscript (Line 205-208).
  • Line 200 Table 2 Nonparametric statistical tests such as the Chi-square test or T-test should be performed. Pearson's chi-square test was applied. A paragraph for the statistical analysis is added in the Material and methods section, and the result of the Chi-square test was written in the result and discussion section. 
  • Line 224 Reference should be added. A reference is added [42; Ceelen et al, 2006]. 
  • Line 226-227 H. pullorum was not detected in the environment swabs but the “others” do. Two H. pylori were detected in two cutting board samples, while only one H. pullorum was detected in one of the worker hand samples. This is clearly written in the text.
  • Line 229-231 The result is not detailed enough to sustain this argument unless statistical analysis or molecular traceability were performed. The sentence of this argument is deleted from the revised text.
  • Line 260-261 “It was also found that these two H. pullorum isolates of the current study……” which two? The selected five Helicobacter species (3 from H. pylori & 2 from H. pullorum) were stated in the text as ell as in Table 3.  
  • Fig.4 The strains in the current study should be emphasized to get a good understanding of the results. In figure 4, a red circular shape is inserted indicating the three sequenced 16S rRNA of H. pylori; and a blue rhombus shape is inserted to distinguish the two sequenced 16S rRNA of H. pullorum. A better resolution figure substituted the old one.
  • Line 280 Phylogeny analysis and discussion should also focus on the distance of strains isolated from different sample types. This may provide some information on the cross-contamination mentioned before. A  paragraph concerning this phylogeny analysis in relation to the variation of sampling area is written in the revised manuscript.
  • Line 370-374 The frequent use of antibiotics for prophylaxis and growth promotion in the poultry industry may be the main reason but not the translation from human staff. The paragraph is corrected to elucidate that the misuse and overuse of antimicrobials in the poultry industry is the main source of the emergence of multidrug-resistant foodborne pathogens. 

                Thank you so much for your valuable revision and corrections